# Comparison of laboratory-based and non-laboratory-based WHO and GLOBORISK CVD risk scores: A cross-sectional analysis of the APCAPS cohort

Hemant Mahajan[1]*, Poppy Alice Carson Mallinson[2]*, Judith Lieber[2], Santhi Bhogadi[1], Santosh Kumar Banjara[1], Anoop Shah[2], Vipin Gupta[3], Gagandeep Kaur Walia[4], Bharati Kulkarni[1], Sanjay Kinra[2]

1 Indian Council of Medical Research - National Institute of Nutrition, Hyderabad, India, 2 London School of Hygiene & Tropical Medicine, London, United Kingdom, 3 University of Delhi, Delhi, India, 4 Public Health Foundation of India, Delhi, India

* hemant.mahajan.84@gmail.com (HM); Poppy.Mallinson1@lshtm.ac.uk (PACM)

## Abstract

### Background and Aim

Cardiovascular diseases (CVDs) represent a growing public-health challenge in India, where nearly one in four deaths is CVD-related. Accurate risk stratification underpins targeted prevention, yet laboratory-dependent tools are often impractical in resource-limited settings. The World Health Organization (WHO) and GLOBORISK initiatives both offer non-laboratory-based 10-year CVD risk algorithms alongside their laboratory-based counterparts. We aimed to compare laboratory- and non-laboratory-based WHO and GLOBORISK CVD risk scores, assess their concordance, and examine relationships with sub-clinical atherosclerosis in a rural Indian cohort.

### Materials and Methods

We conducted a cross-sectional analysis of 2,465 adults (1,184 men, 1,281 women) aged 40−74 years from the third wave (2010−12) of the Andhra Pradesh Children and Parents Study (APCAPS). Participants with prior CVD were excluded. Ten-year CVD risk was calculated using sex-specific WHO (South Asia) and India-calibrated GLOBORISK models, both laboratory-based (age, sex, smoking, systolic blood pressure, diabetes, total cholesterol) and non-laboratory-based (age, sex, smoking, systolic blood pressure, BMI) algorithms. Categorical agreement was quantified via percentage agreement and quadratic weighted kappa (κ); continuous agreement by Bland-Altman analysis. We also evaluated linear associations between each risk score (categorical and continuous) and three sub-clinical atherosclerosis markers: carotid intima-media thickness (CIMT), pulse-wave velocity (PWV), and augmentation

**Data availability statement:** Data cannot be shared publicly because of privacy policy of the Indian Council of Medical Research, India. Data are available from the ICMR-NIN Institutional Data Access / Ethics Committee (contact via email: brsham@uohyd.ac.in (Dr. Shamanna BR, Chair-person)) for researchers who meet the criteria for access to confidential data.

**Funding:** This work was supported by The Wellcome Trust, UK (Award Number: 083707, 084774 and 084754) and Medical Research Council, UK (Award Number: MR/T038292/1 and MR/V001221/1), with in-kind support (infrastructure) from the Indian Council of Medical Research – National Institute of Nutrition, India. The funders had no role in study design, data collection and analysis, decision to publish, or preparation of the manuscript.

**Competing interests:** No authors have competing interests.

index (AIx), through sex-stratified multi-level linear regression with random intercept at the household level, adjusting for multiple testing (p < 0.01).

## Results

Median WHO-CVD-risk was 6.0% (IQR 4% − 9%) in men and 3.0% (2% − 4%) in women for both lab and non-lab models; median GLOBORISK-CVD-risk was 12.0% (9% − 16%) for lab-model vs. 15.0% (10% − 16%) for non-lab-model in men and 5.0% (3% − 9%) for lab-model vs. 5.0% (3% − 9%) for non-lab-model in women. Categorical agreement was substantial to almost perfect: WHO κ = 0.82 (overall), GLOBORISK κ = 0.72. Bland-Altman analyses demonstrated mean differences <1% between lab- and non-lab-based scores, though non-lab models underestimated risk by 4.2% in diabetics and 1.2% in participants with total cholesterol ≥200 mg/dL. Both risk scores showed positive, dose-response relationships with CIMT, PWV, and AIx (p-trend<0.001), with each SD increase in CVD-scores associated with clinically meaningful increases in all three markers of sub-clinical atheroscerosis.

## Conclusion

Non-laboratory-based WHO and GLOBORISK CVD risk scores exhibit high overall agreement with laboratory-based models and correlate strongly with subclinical atherosclerosis in rural India. However, modest underestimation in high-risk subgroups (diabetics, hypercholesterolemia) warrants cautious interpretation. These findings support the feasibility of non-lab risk assessment in resource-constrained settings, while underscoring the need for prospective validation against hard cardiovascular outcomes prior to large-scale implementation.

## Introduction

Cardiovascular diseases (CVD) are the leading global cause of death, with a growing burden in developing countries like India. Nearly a quarter of all deaths in India are attributed to CVD, with an age-standardized death rate of 272 per 100,000 population, higher than the global average of 235 per 100,000 [1]. CVD prevalence in India has increased fivefold over three decades, accounting for 26.6% of total deaths and 13.6% of disability-adjusted life years (DALYs) in 2017, a significant rise from 1990 [2,3]. This escalating burden poses significant economic and healthcare challenges, highlighting the need for effective primary prevention strategies.

CVD predicting risk scores such as the Framingham Risk Score (FRS), INTER-HEART Modifiable Risk Score (IHMRS) [4], WHO-CVD risk charts, SCORE, GLO-BORISK, and the Revised Pooled Cohort Equation (RPCE), among others [2,5–10] assess CVD risk by considering the synergistic effects of risk factors like hypertension, diabetes, dyslipidaemia, obesity, and smoking. These cost-effective tools enable personalized risk mitigation through lifestyle modifications (such as increased physical activity, heart-healthy diets, and smoking cessation) and medications

(lipid-lowering, blood pressure-lowering, and anti-thrombotic) to reduce CVD-related morbidity and mortality. However, most of these risk scores are based on data from high-income countries and may not accurately estimate CVD risk for the Indian population due to different risk factor profiles and CVD event rates [2].

Given the lack of an indigenous CVD prediction score in India, GLOBORISK calibrated to Indian data [10] or WHO-CVD risk scores developed for South Asia are commonly used [5]. Both provide non-laboratory-based algorithms (in addition to lab-based algorithm) that rely on easily obtainable variables such as age, sex, smoking status, systolic blood pressure (SBP), and body mass index (BMI), making them suitable for use in resource-limited settings without laboratory infrastructure. These non-laboratory-based algorithms can be integrated into India's Non-Communicable Disease (NCD) solution applications: mobile and tablet-based platforms developed for NCD prevention, screening, and management [11,12]. For example, as part of the Comprehensive Primary Health Care initiative in Telangana, Accredited Social Health Activists (ASHAs) use the state government's 'ASHA Disease Profile' app to capture data on risk factors and symptoms of communicable and non-communicable diseases across the population [13].

Despite the advantages of non-laboratory-based CVD prediction algorithms, their performance relative to laboratory-based models have not been extensively evaluated in Indian populations. Some studies [14–16] have assessed their predictive ability, but limitations such as small sample sizes, lack of sex-stratified analyses, absence of validation using clinical or subclinical CVD markers, reliance on point-of-care devices for blood sugar measurement, and restriction of analysis to individuals aged 30–69 years (despite the WHO-CVD risk score being designed for individuals aged 40–74 years) have constrained their generalizability. Notably, Phillip et al. [17] using data from the PURE study, developed region-specific non-laboratory-based INTERHEART Modifiable Risk Score(IHMRS) models, including one for South Asia. Their findings demonstrated that non-laboratory-based risk scores can perform comparably to laboratory-based models. However, the IHMRS was specifically designed for myocardial infarction (MI) using a case-control study and requires multiple variables-including age, sex, diabetes, hypertension, smoking history, second-hand smoke exposure, physical activity, depression, general stress, waist-to-hip ratio, diet, and meat intake making it more complex than WHO-CVD risk scores or GLOBORISK.

In this study, we aimed to estimate predicted CVD risk in the Andhra Pradesh Children and Parents Study (APCAPS) cohort using the GLOBORISK CVD score recalibrated for India [10] and updated WHO-CVD risk prediction tools [5]. We also sought to assess the concordance between laboratory-based and non-laboratory-based scores, offering insights into their utility in resource-constrained Indian settings.

## Materials and methods

### Study population

The APCAPS is based in 29 rural villages and towns located approximately 50 km from Hyderabad. Initially, APCAPS was established as a long-term follow-up study of the offspring from a pregnancy nutrition trial conducted between 1987 and 1990. Over time, the study expanded to include family members of the original trial participants. For the present analysis, we utilized cross-sectional data from the third wave of data collection (2010-2012), during which 6,944 of the 10,213 (68%) invited family members participated in clinical examinations [18]. Data for the third wave of APCAPS were accessed for the current analysis on 01/01/2025 (PI: Prof. Sanjay Kinra). The study adhered to the ethical principles outlined in the Declaration of Helsinki. Ethical approval was granted by the London School of Hygiene & Tropical Medicine (London, UK) [LSHTM Ethics Ref: 21771], the ICMR-National Institute of Nutrition (Hyderabad, India) [NIN Protocol Number: 04/01/2020], and the Indian Institute of Public Health (Hyderabad, India) [IIPHH/ TRCIEC/189/2018]. Permissions from village heads and governing committees were obtained verbally, and informed written consent (or a witnessed thumbprint for illiterate participants) was collected from all participants prior to their enrolment in the study.

## Clinical assessment

As previously published [18,19], trained interviewers collected data on tobacco and alcohol use at village clinics using standard questions from India's Third National Family Health Survey. Socioeconomic status was assessed with 14 selected questions from the Standard of Living Index (SLI), where higher scores indicated higher socioeconomic status. Weight and height were measured twice using standard protocols, with discrepancies prompting a third reading. BMI was calculated as weight (kg) divided by height (m²). Blood pressure was measured in the right arm using an Omron M5-I oscillometric device after a 5-minute rest. Three readings were taken one minute apart, with the mean of the last two used for analysis. Hypertension was defined as a prior clinical diagnosis and/or SBP ≥ 140 mmHg or diastolic blood pressure (DBP) ≥90 mmHg [18].

Venous blood samples were collected after an 8-hour fast, and glucose was measured using the GOD-PAP method on the same day. Lipids (total cholesterol, triglycerides, high density lipoprotein-cholesterol (HDL-C)) were quantified using enzymatic assays, with quality maintained through internal controls and participation in external quality programs. The Friedewald equation was used to calculate LDL-C [20]. Assays had intra- and inter-assay variation coefficients below 3% and 5%, respectively [18]. Diabetes was defined as a prior clinical diagnosis and/or fasting glucose ≥126 mg/dL. The reliability of clinical measurements was ensured by strict protocols, regular calibration, and monitoring for consistency. Reproducibility, assessed with repeat measurements in a random 5% subsample, showed intraclass correlation coefficients of >0.98 for anthropometry, > 0.85 for vascular measures, and >0.94 for biochemical assays [18].

## CVD-risk prediction scores: WHO and GLOBORISK

The 10-year CVD risk was calculated using both laboratory-based and non-laboratory-based charts for WHO CVD-risk prediction (South Asia) and GLOBORISK (India) for males and females aged 40-74 years, without a self-reported history of CVD (heart disease or coronary bypass surgery or stroke) [5,10]. For both the CVD risk prediction scores (WHO as well as GLOBORISK) the sex-specific laboratory-based model included the variables age, smoking, diabetes, SBP and total cholesterol, while the sex-specific non-laboratory-based model included age, smoking, SBP, and BMI [5]. Analysis was restricted to participants age 40-74 years, and complete data for age, SBP, BMI, fasting glucose, and total cholesterol. The endpoints used to develop both the CVD risk prediction scores were: fatal or non-fatal myocardial infarction or coronary heart disease, and fatal or non-fatal stroke over a 10-year period (Fig 1) [5]. For both the CVD risk prediction scores, an individual is grouped into the 'high' cardiovascular risk with a 10-year risk of ≥20%. 'Low' risk was defined as cardiovascular risk with a 10-year risk of <10%. All other risk values were categorized as the 'intermediate' [5,10,21].

The validity of WHO-CVD and GLOBORISK risk prediction scores was assessed separately for males and females using measures of subclinical atherosclerosis: Carotid Intima-Media Thickness (CIMT), Pulse Wave Velocity (PWV), and Augmentation Index (AIx), which are considered precursors of CVD. Standardized clinical procedures for measuring CIMT, PWV, and AIx have been documented elsewhere [22].

## Statistical analyses

Categorical variables were expressed as numbers (percentages), and continuous variables were presented as means and standard deviations (SDs) or median and inter-quartile range (IQR) depending on the normal distribution of a continuous variable. Linear multilevel modes with random intercept at the household level (to account for the potential correlation of CVD risk factors among the members of the same household) was used separately for males and females to explore the association between WHO and GLOBORISK CVD risk scores and measures of subclinical atherosclerosis. CVD risk scores were considered both as categorical variables (classified into low, intermediate, and high-risk categories) and as continuous variables to assess their relationship with subclinical atherosclerosis markers. To examine trends across CVD risk categories, median values were assigned to each category and treated as a continuous variable. Additionally,

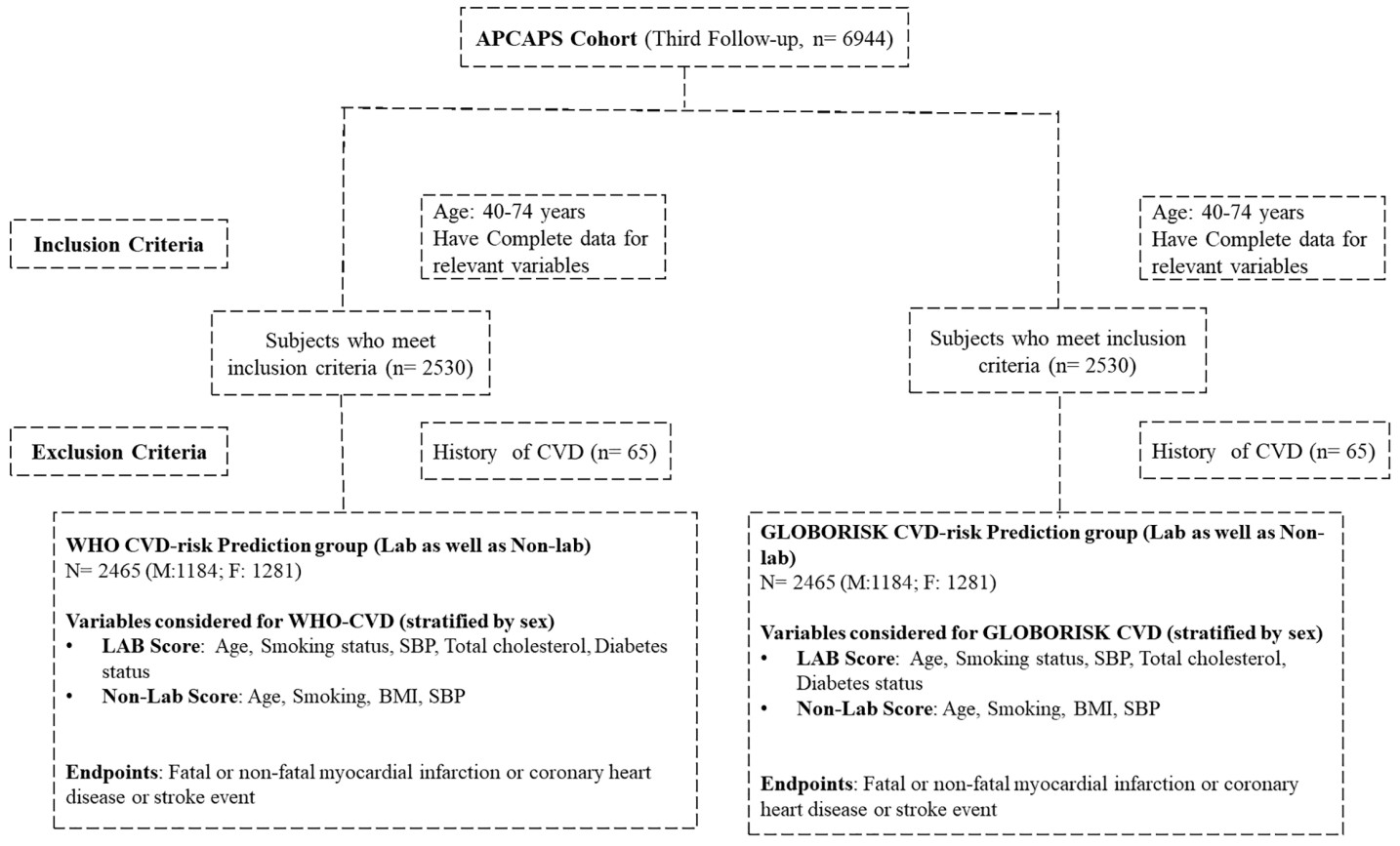

**Fig 1. Description of the study population and each risk score assessment groups.** n, frequency; M, Male; F, Female; SBP, systolic blood pressure; HDL-C, high density lipoprotein-cholesterol; WHO-CVD, World Health Organization cardiovascular disease risk charts; CVD, cardiovascular disease; DM, type 2 diabetes mellitus; BMI, body mass index; MI, myocardial infarction; CHD, coronary heart disease; PAD, peripheral arterial disease;.

the change in subclinical atherosclerosis markers per SD increase in CVD scores was estimated separately for males and females. All p-values were two-tailed. Given that we examined associations of CVD risk scores with three correlated subclinical atherosclerosis markers (CIMT, PWV, Aix), we prespecified a more conservative significance level of p < 0.01 to partially account for multiple testing, (while recognising that a strict Bonferroni correction would be overly conservative due to the non-independence of these outcomes).

Categorical CVD risk agreement was assessed using percent agreement and quadratic weighted kappa statistics [23], while Bland-Altman plots evaluated the agreement between continuous CVD risk scores from laboratory-based and non-laboratory-based models [24]. The Kappa values were classified as poor (≤ 0), slight (0.01–0.20), fair (0.21–0.40), moderate (0.41–0.60), substantial (0.61–0.80), and almost perfect (0.81–1.0) [25]. For Bland-Altman plots, the difference in risk (laboratory-based minus non-laboratory-based) and the mean risk scores were calculated. These plots were stratified by sex, and mean differences with 95% confidence intervals (CIs) were computed using paired t-tests. The plots show the relationship between the difference in scores (vertical axis) and their mean (horizontal axis). Limits of agreement, defined as the mean difference ± 1.96 SD of score differences, were also presented [24]. The quadratic weighted Kappa statistics and Bland-Altman analysis were performed overall, as well as separately for males, females, and participants with specific risk factors, including smoking, diabetes, hypertension, BMI ≥ 25 kg/m², and total cholesterol ≥ 200 mg/dL.

## Results

### CVD-risk score analysis

A total of 1,184 males and 1,281 females were analyzed for the WHO and GLOBORISK CVD-risk score. The mean (SD) age was 53.47 years (6.73) for males and 47.22 years (5.47) for females. Among males, the median (IQR) lab- and non-lab-based WHO CVD-risk scores were both 6.0% (4.0, 9.0), with 22–25% of males having a predicted risk ≥10%. For females, the median (IQR) lab- and non-lab-based WHO CVD-risk score was 3.0% (2.0, 4.0), with 3% having a predicted risk ≥10%. Whereas, for males' median (IQR) lab- and non-lab-based GLOBORISK CVD-risk scores were 12.0 (9.0, 16.0) and 15.0 (10.0, 16.0), respectively. Approximately 70% and 80% males were having lab- and non-lab-based GLOBORISK CVD-risk scores ≥10%. For females, the median (IQR) lab- and non-lab-based GLOBORISK CVD-risk score was 5.0 (3.0, 9.0) with approximately 21% having a predicted lab-based risk and approximately 24% having a predicted non-lab-based risk ≥10% (Table 1).

The mean (SD) values for subclinical atherosclerosis markers were: CIMT was 1.001 mm (0.215), PWV was 8.02 m/s (1.34), and AIx was 29.44% (8.38) in males. Among females, CIMT was 0.985 mm (0.182), PWV was 7.64 m/s (1.24), and AIx was 30.40% (9.11) (Table 1). There was a linear and positive association between CVD-prediction scores (WHO-CVD and GLOBORISK CVD risk scores) and the measures of subclinical atherosclerosis. The levels of CIMT, PWV, and AIx were increasing with the increase in the severity levels of CVD-prediction scores (low, intermediate and high CVD-risk for WHO as well as GLOBORISK, $p_{linear-trend}$ <0.01) (Table 2).

For males, one SD (4.77) increase in WHO Lab-based CVD-Score was associated with 0.048 mm (0.030, 0.067), 0.579 m/s (0.478, 0.679), and 2.23% (1.57, 2.89) higher CIMT, PWV, and AIx, respectively. One SD (4.29) increase in WHO non-Lab-based CVD-Score was associated with 0.048 mm (0.031, 0.065), 0.603 m/s (0.502, 0.705), and 2.34% (1.64, 3.05) higher CIMT, PWV, and AIx, respectively. One SD (7.63) increase in GLOBORISK Lab-based CVD-Score was associated with 0.041 mm (0.021, 0.061), 0.421 m/s (0.322, 0.520), and 2.28% (1.69, 2.86) higher CIMT, PWV, and AIx, respectively. One SD (5.97) increase in GLOBORISK non-Lab-based CVD-Score was associated with 0.041 mm (0.024, 0.058), 0.504 m/s (0.397, 0.611), and 2.54% (1.96, 3.12) higher CIMT, PWV, and AIx, respectively (Table 2).

For females, one SD (2.54) increase in WHO Lab-based CVD-Score was associated with 0.031 mm (0.018, 0.044), 0.338 m/s (0.250, 0.425), and 1.98% (1.41, 2.55) higher CIMT, PWV, and AIx, respectively. One SD (2.61) increase in WHO non-Lab-based CVD-Score was associated with 0.029 mm (0.017, 0.042), 0.314 m/s (0.222, 0.406), and 2.12% (1.48, 2.76) higher CIMT, PWV, and AIx, respectively. One SD (5.75) increase in GLOBORISK Lab-based CVD-Score was associated with 0.029 mm (0.015, 0.042), 0.286 m/s (0.193, 0.378), and 1.80% (1.15, 2.44) higher CIMT, PWV, and AIx, respectively. One SD (5.46) increase in GLOBORISK non-Lab-based CVD-Score was associated with 0.026 mm (0.014, 0.038), 0.202 m/s (0.117, 0.286), and 1.81% (1.23, 2.38) higher CIMT, PWV, and AIx, respectively (Table 2).

### Categorical agreement and kappa coefficients

For the WHO CVD risk scores, the percentage agreement between risk categories was 98.6% with a quadratic weighted kappa (95% CI) of 0.82 (0.78, 0.85). The quadratic weighted kappa (95% CI) among males and females were 0.81 (0.77, 0.85) and 0.73 (0.62, 0.84), respectively. When analysed across different risk factors, quadratic weighted kappa was substantial to almost perfect except among participants diagnosed with diabetes. The percentage agreement between risk categories was 91.0% and the quadratic weighted kappa (95% CI) was 0.53 (0.42, 0.64) among participants diagnosed with diabetes (Table 3).

For the GLOBORISK CVD risk scores, the percentage agreement between risk categories was 94.0% with a quadratic weighted kappa (95% CI) of 0.72 (0.69, 0.74). The quadratic weighted kappa (95% CI) among males and females were 0.63 (0.59, 0.67) and 0.63 (0.58, 0.68), respectively. When analysed across different risk factors, quadratic weighted kappa was moderate to substantial except among participants diagnosed with diabetes and among smokers. The

**Table 1. Characteristics of APCAPS participants (follow-up 3) considered for WHO- and GLOBORISK-CVD-risk prediction scores calculation (n = 2465).**

| Variables | Male (n = 1184) | Female (n = 1281) |
|---|---|---|
| Age in years, mean (SD) | 53.47 (6.73) | 47.22 (5.47) |
| Smoking, n (%) | 710 (60.0) | 314 (24.5) |
| BMI in kg/m², mean (SD) | 20.50 (3.67) | 21.87 (4.14) |
| HDL-C in mg/dL, mean (SD) | 45.2 (15.1) | 44.8 (13.1) |
| Total Cholesterol mg/dL, mean (SD) | 141.1 (87.4) | 179.8 (37.5) |
| Systolic Blood Pressure in mmHg, mean (SD) | 128.69 (19.34) | 122.29 (16.15) |
| Antihypertensive medications, n (%) | 87 (7.4) | 70 (5.5) |
| Fasting glucose in mg/dL, mean (SD) | 99.3 (27.9) | 98.0 (26.7) |
| Diabetes mellitus, n (%) | 88 (7.4) | 75 (5.9) |
| *WHO Lab-based predicted CVD risk, median (IQR)* | *6.0 (4.0, 9.0)* | *3 (2.0, 4.0)* |
| - Low, n (%) | 892 (75.3) | 1240 (96.8) |
| - Intermediate, n (%) | 252 (21.3) | 39 (3.0) |
| - High, n (%) | 40 (3.4) | 2 (0.2) |
| *WHO non-Lab-based predicted CVD risk, median (IQR)* | *6.0 (4.0, 9.0)* | *3 (2.0, 4.0)* |
| - Low, n (%) | 913 (77.1) | 1241 (96.9) |
| - Intermediate, n (%) | 250 (21.1) | 39 (3.0) |
| - High, n (%) | 21 (1.8) | 1 (0.1) |
| *GLOBORISK Lab-based predicted CVD risk, median (IQR)* | *12.0 (9.0, 16.0)* | *5.0 (3.0, 9.0)* |
| - Low, n (%) | 353 (29.8) | 1018 (79.5) |
| - Intermediate, n (%) | 661 (55.8) | 220 (17.2) |
| - High, n (%) | 170 (14.4) | 43 (3.4) |
| *GLOBORISK non-Lab-based predicted CVD risk, median (IQR)* | *15.0 (10.0, 16.0)* | *5.0 (3.0, 9.0)* |
| - Low, n (%) | 233 (19.7) | 973 (76.0) |
| - Intermediate, n (%) | 788 (66.6) | 253 (19.8) |
| - High, n (%) | 163 (13.8) | 55 (4.3) |
| *Subclinical-atherosclerosis biomarkers* | | |
| - CIMT in mm, mean (SD) | 1.001 (0.215) | 0.985 (0.182) |
| - PWV in m/s, mean (SD) | 8.02 (1.34) | 7.64 (1.24) |
| - Augmentation Index (%), mean (SD) | 29.44 (8.38) | 30.40 (9.11) |

%, Percentage; APCAPS, Andhra Pradesh Children and Parents Study; SD, standard deviation; IQR, interquartile range; CIMT, carotid intima media thickness; PWV, pulse wave velocity; and AIx, Augmentation Index; SD, standard deviation; WHO, world health organization; CVD, cardiovascular diseases; mm, millimeter; m/s, meter per second; HDL-C, high density lipoprotein cholesterol;

For CIMT, PWV, and AIx, data was available for 723, 715, and 697 for males and 836, 843, and 813 for females, respectively;

quadratic weighted kappa (95% CI) was 0.42 (0.35, 0.49) among participants diagnosed with diabetes and 0.48 (0.43 to 0.52) among smokers (Table 3).

## Bland-Altman plots: mean differences and limits of agreement

For the WHO CVD risk scores, the mean difference between lab-based and non-lab-based scores was 0.10 (0.03, 0.17). The limits of agreement (lower and upper) were −31.9 and 3.39, respectively. Among males and females, the mean difference between lab-based and non-lab-based scores were 0.31% (95% CI: 0.19, 0.43) and −0.09% (95% CI: −0.15, −0.03), respectively. The limits of agreement (lower and upper) were −3.87% to 4.49% for males and −2.19% to 2.01% for

**Table 2. Levels of CIMT, PWV and AIx across the categories of WHO-CVD and GLOBORISK scores as well as β coeff. (95% CI) for CIMT, PWV and AIx per SD change in CVD scores.**

**Carotid Intima Media Thickness (in mm)**

|  | CVD Risk Categories | | | p-trend | β Coeff. (95% CI) per SD change in CVD score |
|---|---|---|---|---|---|
|  | Low | Intermediate | High | | |
| *Males* | | | | | |
| WHO Lab-based CVD- Score | 0.975 (0.959, 0.991) | 1.076 (1.036, 1.117) | 1.118 (0.995, 1.241) | 0.024 | 0.048 (0.030, 0.067) |
| WHO non-Lab-based CVD- Score | 0.977 (0.960, 0.995) | 1.079 (1.042, 1.115) | 1.127 (1.006, 1.248) | 0.016 | 0.048 (0.031, 0.065) |
| GLOBORISK Lab-based CVD- Score | 0.940 (0.917, 0.963) | 1.015 (0.995, 1.035) | 1.061 (1.006, 1.115) | <0.001 | 0.041 (0.021, 0.061) |
| GLOBORISK non-Lab-based CVD- Score | 0.929 (0.896, 0.963) | 0.999 (0.982, 1.017) | 1.098 (1.045, 1.151) | <0.001 | 0.041 (0.024, 0.058) |
| Females | | | | | |
| WHO Lab-based CVD- Score | 0.983 (0.970, 0.995) | 1.062 (0.981, 1.144) | 1.11 (0.755, 1.466) | <0.001 | 0.031 (0.018, 0.044) |
| WHO non-Lab-based CVD- Score | 0.984 (0.971, 0.997) | 1.024 (0.950, 1.010) | 1.11 (0.753, 1.467) | <0.001 | 0.029 (0.017, 0.042) |
| GLOBORISK Lab-based CVD- Score | 0.978 (0.964, 0.992) | 1.012 (0.985, 1.038) | 1.042 (0.955, 1.130) | 0.152 | 0.029 (0.015, 0.042) |
| GLOBORISK non-Lab-based CVD- Score | 0.970 (0.957, 0.984) | 1.035 (1.005, 1.066) | 1.016 (0.956, 1.077) | 0.143 | 0.026 (0.014, 0.038) |

**Pulse Wave Velocity (in m/s)**

|  | CVD Risk Categories | | | p-trend | β Coeff. (95% CI) per SD change in CVD score |
|---|---|---|---|---|---|
|  | Low | Intermediate | High | | |
| *Males* | | | | | |
| WHO Lab-based CVD- Score | 7.71 (7.62, 7.81) | 8.88 (8.64, 9.11) | 9.35 (8.71, 9.99) | <0.001 | 0.579 (0.478, 0.679) |
| WHO non-Lab-based CVD- Score | 7.74 (7.64, 7.84) | 8.93 (8.69, 9.16) | 10.10 (9.15, 11.04) | <0.001 | 0.603 (0.502, 0.705) |
| GLOBORISK Lab-based CVD- Score | 7.64 (7.49, 7.79) | 7.95 (7.82, 8.08) | 8.94 (8.69, 9.20) | <0.001 | 0.421 (0.322, 0.520) |
| GLOBORISK non-Lab-based CVD- Score | 7.55 (7.37, 7.74) | 7.89 (7.78, 8.00) | 9.27 (8.97, 9.57) | <0.001 | 0.504 (0.397, 0.611) |
| *Females* | | | | | |
| WHO Lab-based CVD- Score | 7.61 (7.53, 7.70) | 8.41 (7.86, 8.96) | 9.37 (8.35, 10.38) | <0.001 | 0.338 (0.250, 0.425) |
| WHO non-Lab-based CVD- Score | 7.61 (7.52, 7.69) | 8.54 (8.01, 9.07) | 8.63 (6.21, 11.06) | <0.001 | 0.314 (0.222, 0.406) |
| GLOBORISK Lab-based CVD- Score | 7.53 (7.44, 7.63) | 7.97 (7.76, 8.18) | 8.58 (7.89, 9.26) | 0.003 | 0.286 (0.193, 0.378) |
| GLOBORISK non-Lab-based CVD- Score | 7.57 (7.48, 7.67) | 7.73 (7.52, 7.93) | 8.30 (7.89, 8.72) | <0.001 | 0.202 (0.117, 0.286) |

**Augmentation Index (in %)**

|  | CVD Risk Categories | | | p-trend | β Coeff. (95% CI) per SD change in CVD score |
|---|---|---|---|---|---|
|  | Low | Intermediate | High | | |
| *Males* | | | | | |
| WHO Lab-based CVD- Score | 28.34 (27.65, 29.04) | 33.16 (31.86, 34.46) | 30.70 (26.96, 34.44) | 0.221 | 2.23 (1.57, 2.89) |
| WHO non-Lab-based CVD- Score | 28.42 (27.74, 29.10) | 33.07 (31.68, 34.46) | 32.83 (27.64, 38.03) | 0.097 | 2.34 (1.64, 3.05) |
| GLOBORISK Lab-based CVD- Score | 27.00 (25.89, 28.10) | 29.46 (28.66, 30.26) | 33.98 (32.36, 35.59) | <0.001 | 2.28 (1.69, 2.86) |
| GLOBORISK non-Lab-based CVD- Score | 26.44 (25.04, 27.84) | 29.24 (28.52, 29.97) | 34.42 (32.73, 36.11) | <0.001 | 2.54 (1.96, 3.12) |
| *Females* | | | | | |
| WHO Lab-based CVD- Score | 30.27 (29.63, 30.91) | 34.38 (30.56, 38.19) | 38.83 (33.97, 43.70) | <0.001 | 1.98 (1.41, 2.55) |
| WHO non-Lab-based CVD- Score | 30.25 (29.61, 30.89) | 35.84 (32.96, 38.72) | 35.33 (35.33, 35.33) | <0.001 | 2.12 (1.48, 2.76) |
| GLOBORISK Lab-based CVD- Score | 29.64 (28.94, 30.35) | 33.43 (32.11, 34.76) | 34.35 (30.29, 38.41) | 0.025 | 1.80 (1.15, 2.44) |
| GLOBORISK non-Lab-based CVD- Score | 29.69 (28.95, 30.43) | 32.30 (31.07, 33.53) | 34.56 (32.20, 36.91) | <0.001 | 1.81 (1.23, 2.38) |

β coeff, β coefficient; CI, confidence interval; mm, millimeter; m/s, meter per second; %, Percentage; CIMT, carotid intima media thickness; PWV, pulse wave velocity; and Aix, Augmentation Index; SD, standard deviation; WHO, world health organization; CVD, cardiovascular diseases;

Levels of CIMT, PWV, and AIx are expressed as mean (95% CI) across the categories of CVD-scores (low, intermediate, and high);

• For *males* One SD of WHO Lab-based CVD-Score, WHO non-Lab-based CVD-Score, GLOBORISK Lab-based CVD-Score, GLOBORISK non-Lab-based CVD-Score equal to 4.77, 4.29, 7.63, and 5.97, respectively;

• For *females* One SD of WHO Lab-based CVD-Score, WHO non-Lab-based CVD-Score, GLOBORISK Lab-based CVD-Score, GLOBORISK non-Lab-based CVD-Score equal to 2.54, 2.61, 5.75, and 5.46, respectively;

**Table 3. Agreement between the lab- and non-lab-based WHO- and GLOBORISK-CVD-risk prediction scores.**

| | | WHO CVD Risk Scores | | GLOBORISK CVD Risk Scores | |
|---|---|---|---|---|---|
| | | Percent Agreement (95% CI) | Quadratic Weighted Kappa (95% CI) | Percent Agreement (95% CI) | Quadratic Weighted Kappa (95% CI) |
| Overall (n = 2465) | | 98.6 (98.4 - 98.9) | 0.82 (0.78 - 0.85) | 94.0 (93.5 - 94.5) | 0.72 (0.69 - 0.74) |
| Sex | Men (n = 1184) | 97.6 (97.2 - 98.1) | 0.81 (0.77 - 0.85) | 92.9 (92.2 - 93.6) | 0.63 (0.59 - 0.67) |
| | Women (n = 1281) | 99.5 (99.4 - 99.7) | 0.73 (0.62 - 0.84) | 95.0 (94.3 - 95.6) | 0.63 (0.58 - 0.68) |
| Smoker | Yes (n = 1024) | 97.4 (96.9 - 98.0) | 0.81 (0.77 - 0.85) | 92.0 (91.3 - 92.7) | 0.48 (0.43 - 0.52) |
| | No (n = 1441) | 99.5 (99.3 - 99.7) | 0.66 (0.56 - 0.77) | 95.4 (94.8 - 96.1) | 0.55 (0.49 - 0.61) |
| Diabetes | Yes (n = 184) | 91.0 (88.7 - 93.4) | 0.53 (0.42 - 0.64) | 78.0 (74.6 - 81.4) | 0.42 (0.35 - 0.49) |
| | No (n = 2281) | 99.2 (99.1 - 99.4) | 0.88 (0.85 - 0.91) | 95.3 (94.9 - 95.7) | 0.76 (0.74 - 0.78) |
| SBP | ≥140 mmHg (n = 472) | 96.5 (95.6 - 97.4) | 0.80 (0.75 - 0.85) | 90.5 (89.2 - 91.8) | 0.68 (0.63 - 0.73) |
| | <140 mmHg (n = 1993) | 99.1 (98.9 - 99.4) | 0.77 (0.71 - 0.83) | 94.8 (94.3 - 95.3) | 0.65 (0.62 - 0.68) |
| Total cholesterol | ≥200 mg/dL (n = 608) | 97.3 (96.5 - 98.1) | 0.76 (0.67 - 0.82) | 93.6 (92.4 - 94.9) | 0.75 (0.70 - 0.80) |
| | <200 mg/dL (n = 1857) | 99.0 (98.8 - 99.3) | 0.85 (0.81 - 0.88) | 94.1 (93.6 - 94.6) | 0.70 (0.68 - 0.73) |
| BMI | ≥25 kg/m$^2$ (n = 399) | 97.7 (96.9 - 98.6) | 0.74 (0.65 - 0.83) | 92.2 (90.8 - 93.7) | 0.69 (0.63 - 0.75) |
| | <25 kg/ m$^2$ (n = 2066) | 98.8 (98.5 - 99.1) | 0.83 (0.80 - 0.87) | 94.3 (93.8 - 94.8) | 0.72 (0.70 - 0.75) |

CI, confidence interval; SBP, systolic blood pressure; BMI, body mass index;

females (Table 4, S1A, S1B, S1C Figs). When analysed across different risk factors, the mean difference between lab-based and non-lab-based scores was less than 1% except participants diagnosed with diabetes [mean difference (95% CI): 4.17% (3.76%, 4.58%)] and participants with total cholesterol ≥200 mg/dL [mean difference (95% CI): 1.16% (0.98%, 1.34%)].

For the GLOBORISK CVD risk scores, the mean difference between lab-based and non-lab-based scores was −0.53 (−0.70, −0.36). The limits of agreement (lower and upper) were −9.15 and 8.09, respectively. Among males and females, the mean difference between lab-based and non-lab-based scores were −0.66% (95% CI: −0.95%, −0.37%) and −0.41% (95% CI: −0.61%, −0.21%), respectively. The limits of agreement (lower and upper) were −10.62% to 9.30% for males and −7.56% to 6.74% for females (Table 4, S2A, S2B, S2C Figs). When analysed across different risk factors, the mean difference between lab-based and non-lab-based scores was less than 1% except for smokers [mean difference (95% CI): −1.15% (−1.48%, −0.82%)], participants diagnosed with diabetes [mean difference (95% CI): 10.44% (9.38%, 11.50%)] and participants with total cholesterol ≥200 mg/dL [mean difference (95% CI): 2.85% (2.38%, 3.32%)].

## Discussion

The cross-sectional analysis from the APCAPS third follow-up estimated the 10-year CVD risk using both laboratory-based and non-laboratory-based algorithms among participants aged 40–74 years residing in southern India without a prior CVD diagnosis. There was substantial to almost perfect agreement (72% − 82%) between lab-based and non-lab-based risk scores for both models, with consistent agreement observed among men and women (63% − 81%). Bland-Altman analysis showed that the mean difference between the two methods was less than 1% for both WHO and GLOBORISK CVD risk assessments, indicating strong concordance. However, the level of agreement varied across specific risk factor subgroups such as participants diagnosed with diabetes mellitus or participants with total cholesterol levels ≥200 mg/dL. Importantly, both WHO and GLOBORISK scores demonstrated consistent positive, dose-response relationships with all three subclinical atherosclerosis markers: CIMT, PWV and AIx. Individuals classified in higher 10-year risk categories had thicker carotid intima-media, greater arterial stiffness and higher augmentation index than those in the low-risk category, and each standard deviation increase in CVD risk score was associated with higher levels of these vascular

**Table 4. Mean difference and limits of agreement for the Bland-Altman analysis.**

| CVD-prediction scores | Mean Difference (95% CI) | Limit of agreement | | n (%) observations out of LOA |
| --- | --- | --- | --- | --- |
| | | Lower (95% CI) | Upper (95% CI) | |
| **WHO- CVD-Risk Prediction Score** | | | | |
| Overall (n = 2465) | 0.10 (0.03, 0.17) | −3.19 (−3.31, −3.08) | 3.39 (3.28, 3.51) | 111 (4.50) |
| Male (n = 1184) | 0.31 (0.19, 0.43) | −3.87 (−4.08, −3.66) | 4.49 (4.28, 4.70) | 71 (6.00) |
| Female (n = 1281) | −0.09 (−0.15, −0.03) | −2.19 (−2.29, −2.09) | 2.01 (1.91, 2.11) | 42 (3.28) |
| Smokers (n = 1024) | 0.04 (−0.09, 0.17) | −4.19 (−4.42, −3.96) | 4.27 (4.04, 4.50) | 54 (5.27) |
| Diabetes (n = 184) | 4.17 (3.76, 4.58) | −1.34 (−2.04, −0.63) | 9.68 (8.97, 10.38) | 12 (6.52) |
| SBP ≥ 140 mmHg (n = 472) | 0.09 (−0.15, 0.33) | −5.16 (−5.58, −4.74) | 5.34 (4.92, 5.76) | 25 (5.30) |
| Total Cholesterol ≥200 mg/dL (n = 608) | 1.16 (0.98, 1.34) | −3.37 (−3.69, −3.05) | 5.69 (5.37, 6.01) | 35 (5.76) |
| BMI ≥ 25 kg/m² (n = 399) | −0.04 (−0.26, 0.18) | −4.37 (−4.75, −4.00) | 4.29 (3.92, 4.67) | 20 (5.01) |
| **GLOBORISK- CVD-Risk Prediction Score** | | | | |
| Overall (n = 2465) | −0.53 (−0.70, −0.36) | −9.15 (−9.46, −8.85) | 8.09 (7.79,8.40) | 103 (4.18) |
| Male (n = 1184) | −0.66 (−0.95, −0.37) | −10.62 (−11.12, −10.11) | 9.30 (8.80, 9.80) | 52 (4.39) |
| Female (n = 1281) | −0.41 (−0.61, −0.21) | −7.56 (−7.91, −7.22) | 6.74 (6.40, 7.09) | 57 (4.45) |
| Smokers (n = 1024) | −1.15 (−1.48, −0.82) | −11.79 (−12.37, −11.22) | 9.50 (8.92, 10.07) | 48 (4.69) |
| Diabetes (n = 184) | 10.44 (9.38, 11.50) | −3.89 (−5.72, −2.06) | 24.77 (22.94, 26.60) | 10 (5.43) |
| SBP ≥ 140 mmHg (n = 472) | −0.21 (−0.82, 0.40) | −13.40 (−14.45, −12.35) | 12.98 (11.93, 14.03) | 27 (5.72) |
| Total Cholesterol ≥200 mg/dL (n = 608) | 2.85 (2.38, 3.32) | −8.85 (−9.67, −8.03) | 14.55 (13.73, 15.37) | 37 (6.09) |
| BMI ≥ 25 kg/m² (n = 399) | −0.005 (−0.59, 0.58) | −11.63 (−12.64, −10.62) | 11.62 (10.61, 12.63) | 23 (5.76) |

n, frequency; %, percentage; CI, confidence interval; SBP, systolic blood pressure; BMI, body mass index;

markers. This pattern was observed for both laboratory- and non-laboratory-based algorithms and in both sexes, reinforcing the construct validity of these tools in this rural South Asian cohort. As CIMT, PWV and AIx are recognised precursors of overt cardiovascular events, the alignment of higher predicted risk with a more adverse subclinical vascular profile suggests that these scores are capturing biologically meaningful cardiovascular risk, even in the absence of prospective event data.

Our finding of substantial to almost perfect agreement between lab-based and non-lab-based risk scores aligns with previous studies conducted in India [14–16], as well as research by Bendera et al. in Eastern Sub-Saharan Africa [26], Dehghan et al. and Rezaei et al. in Iran [27,28], Hutton-Mensah KA et al. in West Africa [29] and Chhezom et al. in Bhutan [30], validating the effectiveness of non-laboratory-based methods in resource-limited settings. Furthermore, Phillip et al. [17] using data from the PURE study spanning seven distinct global regions, demonstrated that non-laboratory-based tools exhibit predictive accuracy comparable to laboratory-based methods. Collectively, these findings reinforce the reliability of simplified, non-lab-based models for CVD risk assessment across diverse populations. Consistent with previous research, our results support the broader applicability of these models, particularly in settings with limited laboratory resources or access (especially among non-diabetics).

Although the agreement between lab-based and non-lab-based CVD risk scores was substantial, Bland-Altman analysis revealed systematic bias, with non-laboratory-based methods consistently underestimating or overestimating risk compared to laboratory-based models. For WHO-CVD, the mean difference (95% CI) was 0.10 (0.03, 0.17), while for GLOBORISK, it was −0.53 (−0.70, −0.36). Despite the mean difference being less than 1%, the line of equality fell outside the confidence interval in all cases, indicating a persistent bias. Additionally, the variability of differences between the two methods increased with higher risk estimates (observed for both WHO-CVD and GLOBORISK CVD scores, S1A, S1B, S1C, S2A, S2B, S2C Figs). To further assess this trend, we plotted the proportional difference [(Lab-based Score

- Non-lab-based Score)/Mean×100] against the mean of the two measurements. However, systematic bias persisted across all levels of risk estimation, suggesting that non-laboratory-based models tend to underestimate CVD risk, particularly in individuals with intermediate-to-high risk profiles. The extent of bias was notably greater in participants with total cholesterol ≥200 mg/dL (1% − 3%) and in those with diabetes (5% − 11%). Kappa analysis further indicated only moderate concordance (< 50%) between lab-based and non-lab-based models in diabetic individuals, consistent with previous studies [5,10,16]. Bland–Altman analysis confirmed that non-lab-based scores systematically underestimated CVD risk in those with diabetes, especially among individuals with higher overall risk levels (S3A, S3B Figs). This underestimation raises concerns that high-risk individuals classified by lab-based methods may be downgraded to intermediate-risk when assessed using non-lab-based models, potentially leading to suboptimal clinical management.

These findings highlight the need to incorporate diabetes status into CVD prediction models to prevent risk misclassification. For example, the Framingham risk score, which includes self-reported diabetes demonstrated substantial agreement among participants diagnosed with diabetes [16]. While integrating self-reported diabetes may enhance predictive accuracy (overall as well as for participants with diabetes), further prospective validation using cardiovascular event data is essential. Clinicians should exercise caution when applying non-lab-based WHO-CVD score and GLOBORISK CVD-score in populations with diabetes to enhance their accuracy and clinical utility and avoid risk underestimation. Given the varying levels of agreement across different risk factor profiles, rigorous validation using prospective cohort study is crucial before incorporating non-lab-based models into healthcare systems to minimize the risk of CVD mismanagement.

The large disparity in predicted CVD risk between men and women in our cohort warrants comment. Men were on average older and far more likely to smoke than women, with slightly higher systolic blood pressure, whereas women had somewhat higher BMI and total cholesterol. Given that both WHO and GLOBORISK charts use sex-specific coefficients and baseline hazards derived from populations in which age-specific CVD incidence is consistently higher in men, it is expected that the same constellation of risk factors will translate into higher absolute 10-year risk in men. At the same time, we observed strong, graded associations between both WHO and GLOBORISK scores and subclinical atherosclerosis markers (CIMT, PWV and AIx) within each sex, indicating that these tools capture meaningful variation in vascular burden among women as well as men. Together, these findings suggest that while absolute risk estimates should not be compared uncritically across sexes, sex-stratified use of these scores remains informative for identifying higher-risk individuals within each sex in rural South Asian settings. Thus, observed sex differences in absolute predicted risk largely reflected underlying risk-factor distributions and sex-specific model calibration, but dose-response relationships with subclinical atherosclerosis were consistent in both men and women.

Although WHO and GLOBORISK algorithms use similar predictors and were both developed for combined fatal and non-fatal myocardial infarction and stroke, they produced notably different absolute risk estimates in this rural Indian cohort. WHO charts calibrated to South Asia yielded lower median risks and a much smaller proportion of adults, particularly men, with predicted 10-year risk ≥10%, whereas the India-calibrated GLOBORISK equations classified around three-quarters of men and one-quarter of women above this threshold. This discrepancy likely reflects differences in calibration: WHO models are anchored to region-level incidence and average risk-factor distributions, while GLOBORISK explicitly incorporates India-specific age- and sex-specific CVD mortality and national risk-factor profiles. In our data, however, both tools showed similar, graded associations with CIMT, PWV and AIx, suggesting that they rank individuals' underlying CVD risk comparably but differ in the absolute level of predicted risk. These findings underscore that the choice between WHO and GLOBORISK charts in India is not neutral from a policy perspective, as GLOBORISK will identify substantially more individuals for treatment intensification; prospective validation against hard CVD outcomes in Indian populations is needed to determine which calibration is most appropriate for national guidelines.

The study has several notable strengths. First, the large sample size provided sufficient statistical power to calculate CVD risk scores using multiple models and to categorize participants into different risk groups (low, intermediate, and high). Second, the study included a comprehensive range of clinical assessments spanning anthropometric, biochemical,

and cardiovascular parameters conducted using rigorous protocols to ensure high data quality. The use of standardized methods, such as the GOD-PAP assay for glucose and the Omron M5-I device for blood pressure measurement, enhances the reliability of the findings and facilitates comparability with other studies. Third, the study prioritized reproducibility, reporting high intraclass correlation coefficients for key variables, thereby reinforcing confidence in the reliability of the collected data. Fourth, the relationship between WHO- and GLOBORISK-CVD risk models and various measures of subclinical atherosclerosis was evaluated to explore the utility of these models in identifying individuals at high risk for CVD. While this approach is not the gold standard for validating CVD risk prediction models, it provides a degree of confidence in their applicability. Fifth, the analysis employed robust statistical methods, including quadratic weighted kappa statistics and Bland-Altman analysis, to assess agreement between models.

Limitations of the study include: Although the study is based on a large cohort, the rural, South Asian population might limit the generalizability of the findings to other populations or urban settings. Some important variables, such as smoking and alcohol use, are self-reported, which can introduce recall bias or social desirability bias. We excluded participants with missing values for relevant covariates from the analysis and we are not sure about this exclusion will introduce potential biases in the analysis. Although APCAPS participants are clustered within 29 villages, we did not apply a formal survey design with sampling weights and cluster-adjusted standard errors because the cohort was not selected as a probability sample to represent a wider population and our primary aim was to assess internal associations. This may have led to some underestimation of standard errors and slightly narrower confidence intervals. However, the observed associations between CVD risk scores and subclinical atherosclerosis markers were consistent in direction and magnitude across models and highly statistically significant (most p-values <0.001), suggesting that any such bias is unlikely to materially affect our conclusions.

Public health significance – Our findings have important implications for CVD prevention in India, where laboratory testing for diabetes and lipids remains unevenly available because of cost, logistics, and health-system capacity. In such settings, non-laboratory-based CVD risk algorithms built on readily obtainable parameters such as age, blood pressure, BMI, and smoking offer a practical approach for population-level risk stratification and treatment triage. Embedding these tools within routine primary-care and community screening (e.g., under NPCDCS, through ASHAs/ANMs and related digital screening platforms) could help frontline workers identify high-risk adults earlier, prioritize referrals, and support more targeted initiation or intensification of preventive therapy. In turn, scalable, low-cost risk assessment can strengthen equitable delivery of CVD prevention services and inform programmatic resource allocation.

At the same time, our results highlight that "non-laboratory vs laboratory" is only one policy decision. Even when non-laboratory algorithms perform as acceptable substitutes for laboratory versions within a model, the choice of model (WHO vs India-calibrated GLOBORISK) is not neutral: in our setting, adopting GLOBORISK rather than WHO charts would classify substantially more adults particularly men above commonly used treatment thresholds, with major implications for medication eligibility, service demand, and health-system costs. Because this analysis is cross-sectional and does not include prospective CVD outcomes, we cannot determine which calibration is most appropriate for India; therefore, head-to-head validation against hard CVD events is essential before national guidelines select a single model and risk threshold for large-scale implementation.

## Conclusion

Using WHO and GLOBORISK CVD risk prediction models, we calculated 10-year CVD risk in an Indian population and observed high overall concordance between laboratory-based and non-laboratory-based assessments. However, the presence of systematic bias particularly in individuals with diabetes and elevated cholesterol levels underscores the need for prospective validation with cardiovascular event data before these models can be fully integrated into clinical practice in resource-limited settings. The strong, dose-response correlations between both WHO and GLOBORISK risk scores and markers of subclinical atherosclerosis (CIMT, PWV, and AIx) further support their construct validity and potential utility for CVD risk stratification in rural South Asian populations.

## Institutional review board statement

The study was carried out in accordance with the principles of the Helsinki Declaration. The study protocol and tools have been approved by the ethics committees of ICMR-NIN (Application No. CR/1/V/2023; date: 21 October 2021) and Indian Institute of Public Health Hyderabad (IIPHH/TRCIEC/189/2018; date: 20 May 2020), India, and the London School of Hygiene and Tropical Medicine, UK [Application No. 21771/RR/19113; date: 18 March 2022].

## Informed consent statement

We obtained verbal permission from the heads and governing committees of the villages. Written informed consent (or witnessed thumbprint if illiterate) for inclusion in the study was obtained from each participant prior to enrolment.

## Supporting information

**S1A Fig. Bland and Altman plots for all participants: WHO CVD score.**
(TIF)

**S1B Fig. Bland and Altman plots for Men: WHO CVD score.**
(TIF)

**S1C Fig. Bland and Altman plots for Women: WHO CVD score.**
(TIF)

**S2A Fig. Bland and Altman plots for all participants: GLOBORISK CVD score.**
(TIF)

**S2B Fig. Bland and Altman plots for Men: GLOBORISK CVD score.**
(TIF)

**S2C Fig. Bland and Altman plots for Women: GLOBORISK CVD score.**
(TIF)

**S3A Fig. Bland and Altman Plot for people diagnosed with Diabetes: WHO CVD risk scores.**
(TIF)

**S3B Fig. Bland and Altman Plot for people diagnosed with Diabetes: GLOBRISK CVD risk scores.**
(TIF)

## Author contributions

**Conceptualization:** Hemant Mahajan, Bharati Kulkarni, Sanjay Kinra.

**Formal analysis:** Hemant Mahajan.

**Funding acquisition:** Sanjay Kinra.

**Investigation:** Sanjay Kinra.

**Methodology:** Hemant Mahajan, Poppy Alice Carson Mallinson, Bharati Kulkarni, Sanjay Kinra.

**Project administration:** Santhi Bhogadi.

**Software:** Hemant Mahajan.

**Validation:** Hemant Mahajan, Poppy Alice Carson Mallinson.

**Writing – original draft:** Hemant Mahajan.

**Writing – review & editing:** Poppy Alice Carson Mallinson, Judith Lieber, Santhi Bhogadi, Santosh Kumar Banjara, Anoop Shah, Vipin Gupta, Gagandeep Kaur Walia, Bharati Kulkarni, Sanjay Kinra.

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
