## [Decision Letter · Decision Letter 0]

1 Dec 2025

Dear Dr. Mahajan,

Thank you for submitting your manuscript to PLOS ONE. After careful consideration, we feel that it has merit but does not fully meet PLOS ONE’s publication criteria as it currently stands. Therefore, we invite you to submit a revised version of the manuscript that addresses the points raised during the review process.

We look forward to receiving your revised manuscript.

Kind regards,

Satish G Patil, PhD

Academic Editor

PLOS ONE

Journal Requirements:

“This work was supported by The Wellcome Trust, UK (grant codes 083707, 084774 and 084754) and Medical Research Council, UK (grant codes MR/T038292/1 and MR/V001221/1) with in-kind support (infrastructure) from the Indian Council of Medical Research - National Institute of Nutrition, India.”

Reviewers' comments:

Reviewer's Responses to Questions

**Comments to the Author**

1. Is the manuscript technically sound, and do the data support the conclusions?

Reviewer #1: Yes

Reviewer #2: Partly

2. Has the statistical analysis been performed appropriately and rigorously?

Reviewer #1: Yes

Reviewer #2: Yes

3. Have the authors made all data underlying the findings in their manuscript fully available?

Reviewer #1: No

Reviewer #2: Yes

4. Is the manuscript presented in an intelligible fashion and written in standard English?

Reviewer #1: Yes

Reviewer #2: Yes

Reviewer #1: Please, see attached document. .

Reviewer #2: 1. There should be comments on the relationship of CVD risk scores with subclinical atherosclerosis parameters measured.

2. There should be comments on the disparity in CVD risk scores between men and women. This cannot be overlooked as the disparity is huge

3. There should also be a comment on WHO vs Globorisk as there are differences and what could have accounted for these differences

4. What is your comment on smoking status and its mean difference on Bland Altman analysis

**Do you want your identity to be public for this peer review?** For information about this choice, including consent withdrawal, please see our Privacy Policy

Reviewer #1: No

Reviewer #2: **Yes:** Kojo Awotwi Hutton-Mensah

---

## [Author Response · Author response to Decision Letter 1]

15 Dec 2025

Title of the manuscript: Comparison of Laboratory-Based and Non-Laboratory-Based Who and GLOBORISK CVD Risk Scores: a Cross-Sectional Analysis of the APCAPS Cohort

Journal Requirements:

RESPONSE: Document formatted in PLOS ONE's style requirements.

RESPONSE: I have corrected the grant numbers. Now the grant information provided in the ‘Funding Information’ and ‘Financial Disclosure’ sections match.

“This work was supported by The Wellcome Trust, UK (Award Number: 083707, 084774 and 084754) and Medical Research Council, UK (Award Number: MR/T038292/1 and MR/V001221/1) with in-kind support (infrastructure) from the Indian Council of Medical Research - National Institute of Nutrition, India. The funders had no role in study design, data collection and analysis, decision to publish, or preparation of the manuscript”

“This work was supported by The Wellcome Trust, UK (grant codes 083707, 084774 and 084754) and Medical Research Council, UK (grant codes MR/T038292/1 and MR/V001221/1) with in-kind support (infrastructure) from the Indian Council of Medical Research - National Institute of Nutrition, India.”

RESPONSE: The statement saying “The funders had no role in study design, data collection and analysis, decision to publish, or preparation of the manuscript" has been added in the revised manuscripts.

RESPONSE: Ethical approval for the study was obtained from the following institutions:

• London School of Hygiene & Tropical Medicine, London, UK [LSHTM Ethics Ref: 21771]

• ICMR–National Institute of Nutrition (NIN), Hyderabad, India [NIN Protocol Number: 04/01/2020]

• Indian Institute of Public Health, Hyderabad, India [IIPHH/TRCIEC/189/2018]

The de-identified dataset cannot be shared publicly due to the privacy policy of the Indian Council of Medical Research (ICMR), which restricts open data sharing to protect participant confidentiality. These restrictions are in place to ensure compliance with national data protection regulations and ethical standards concerning sensitive health information.

Researchers who meet the criteria for accessing confidential data may request access through the Institutional Data Access/Ethics Committee of ICMR-NIN. Requests can be directed to:

Dr. Shamanna BR

Chair-person

Institutional Ethics Committee - ICMR-NIN, Hyderabad, India

Email: brsham@uohyd.ac.in

RESPONSE: Captions for Supporting Information files included at end of manuscript.

RESPONSE: Thank you for the information.

Response to Reviewer Comments

Reviewer #2:

1. There should be comments on the relationship of CVD risk scores with subclinical atherosclerosis parameters measured.

RESPONSE: Thank you for this helpful suggestion. We have now expanded the Results and Discussion to more explicitly describe the associations between the WHO and GLOBORISK CVD risk scores and the subclinical atherosclerosis markers (CIMT, PWV and AIx). In the ‘Results’ section, we clarify that higher WHO and GLOBORISK risk categories were associated with progressively higher mean CIMT, PWV and AIx, and that each standard deviation increases in both laboratory- and non-laboratory-based scores was significantly associated with higher levels of all three markers in both men and women (Table 2). In the ‘Discussion’, we now emphasise that these positive, dose-response relationships between risk scores and subclinical atherosclerosis support the construct validity of both WHO and GLOBORISK algorithms in this rural South Asian cohort, and indicate that individuals classified as being at higher 10-year CVD risk have already accumulated structural and functional vascular changes.

2. There should be comments on the disparity in CVD risk scores between men and women. This cannot be overlooked as the disparity is huge

Response: We agree that the sex differences in predicted CVD risk are striking and have now made these patterns more explicit in both the ‘Results’ and ‘Discussion’. In our cohort, men were older on average (53.5 vs 47.2 years) and substantially more likely to be current smokers (60% vs 24.5%), with slightly higher systolic blood pressure than women. These differences in underlying risk-factor profiles, together with the sex-specific baseline hazards built into both the WHO and GLOBORISK algorithms, contribute to the higher absolute 10-year risk estimates in men.

Specifically, median WHO risk was 6% in men versus 3% in women, and 22-25% of men but only 3% of women had WHO-predicted risk ≥10%. For GLOBORISK, 70-80% of men but ~21-24% of women had a predicted 10-year risk ≥10%, despite similar or lower total cholesterol levels in men. These findings are consistent with the higher age-specific CVD incidence typically observed in men and with the design of the sex-specific charts.

Importantly, when we examined within-sex associations between risk scores and subclinical atherosclerosis markers (CIMT, PWV and AIx), we observed clear, positive dose-response relationships in both men and women, and similar increases in these markers per SD increase in WHO and GLOBORISK risk scores. This suggests that although absolute predicted risk is lower in women, the risk scores still discriminate vascular burden within women in a biologically plausible way. We now discuss these points explicitly in the revised ‘Discussion’ and highlight that absolute risk comparisons between sexes should be interpreted cautiously, while emphasizing the importance of sex-stratified application of these tools in clinical and public-health settings.

3. There should also be a comment on WHO vs Globorisk as there are differences and what could have accounted for these differences

RESPONSE: We agree with the reviewer that the differences between WHO and GLOBORISK estimates merit explicit comment. In our data, GLOBORISK consistently assigned higher 10-year CVD risk than the WHO charts, particularly among men. For example, while the median WHO 10-year risk in men was 6% (IQR 4-9%), only about 22-25% of men had WHO-predicted risk ≥10%; in contrast, the median GLOBORISK risk was 12-15%, and approximately 70-80% of men were classified as having ≥10% 10-year risk. Among women, WHO classified only ~3% as ≥10% risk, whereas GLOBORISK classified around 21-24% as ≥10% risk, despite similar median risks of 3% (WHO) and 5% (GLOBORISK).

These discrepancies largely reflect differences in how the two tools are derived and calibrated. In this analysis we used WHO charts calibrated to the South Asia region, which rely on regional age- and sex-specific incidence of myocardial infarction and stroke and average regional distributions of major risk factors. By contrast, we used the India-calibrated GLOBORISK equations, which are explicitly recalibrated to country-specific age- and sex-specific CVD mortality rates and national risk-factor distributions. In a setting such as India, where CVD mortality is higher than in several neighbouring countries, GLOBORISK therefore “anchors” the baseline hazard to a higher underlying national risk than the South-Asia-wide WHO charts, and consequently yields higher absolute 10-year risk estimates. Conceptually, both scores use very similar predictors (age, smoking, blood pressure, diabetes, total cholesterol or BMI), and both were developed for combined fatal and non-fatal coronary heart disease and stroke outcomes, suggesting that their relative ranking of individuals is driven by comparable risk-factor gradients, whereas the absolute level of predicted risk is more sensitive to the choice of calibration data.

Importantly, despite these differences in absolute levels and the proportion classified as “high risk”, both WHO and GLOBORISK scores showed very similar dose-response associations with subclinical atherosclerosis in our cohort: higher categories and each SD increase in either score were associated with higher CIMT, PWV and AIx in both sexes. This pattern suggests that the two tools discriminate risk in a broadly comparable way but differ mainly in calibration of absolute risk. We now highlight these points in the ‘Discussion’ and caution that, until prospective Indian event data allow formal calibration and head-to-head validation, programmatic decisions about which tool and threshold to adopt should consider that India-calibrated GLOBORISK will identify a substantially larger proportion of adults especially men as eligible for intensified preventive interventions than the South-Asia-calibrated WHO charts.

4. What is your comment on smoking status and its mean difference on Bland Altman analysis

RESPONSE: We thank the reviewer for raising the issue of smoking status in relation to the Bland-Altman analysis. In our stratified analyses, the agreement between laboratory-based and non-laboratory-based scores among smokers remained high for the WHO charts and showed only a modest systematic bias for the GLOBORISK charts. For WHO, the mean difference in smokers was 0.04 percentage points (95% CI: -0.09, 0.17), with limits of agreement from -4.19 to 4.27, indicating virtually no systematic bias between lab- and non-lab-based WHO scores among current smokers. In contrast, for GLOBORISK the mean difference in smokers was -1.15 percentage points (95% CI: -1.48, -0.82), meaning that non-laboratory-based GLOBORISK scores tend to classify smokers at slightly higher 10-year CVD risk than their laboratory-based counterparts, by just over one percentage point on average. Although this difference is statistically significant, its absolute magnitude is small compared with the much larger biases seen in participants with diabetes (mean difference 10.44%) or total cholesterol ≥200 mg/dL (2.85%), and the limits of agreement remain comparable to those observed in the overall sample. Consistent with these findings, categorical agreement among smokers is still substantial for WHO (percent agreement 97.4%, κ=0.81) and moderate for GLOBORISK (percent agreement 92.0%, κ=0.48). Taken together, these results suggest that smoking status does not materially compromise the concordance between lab- and non-lab-based WHO scores, and that for GLOBORISK the non-lab algorithm provides a slightly more conservative (higher) risk estimate in smokers rather than underestimating their risk.

Minor Comments

1. Is this still an on-going study. If no, then it should be “was”

Response: The study is ongoing.

2. Was LDL-C derived?

Response: Yes, using Friedewald’s Formula.

3. Did you consider peripheral artery disease?

Response: Both WHO and GLOBORISK CVD risk Score don’t consider peripheral artery disease.

Was this done (The validity of WHO-CVD and GLOBORISK risk prediction scores) as part of the earlier study in 2010-2012 or this was done by the current authors?

Response: The measurements CIMT, PWV, and Augmentation Index were measured in 2010-12 (APCPAS follow-up 3); The validity of the CVD risk scores against these measures were done in the current anlysis.

REVIEWER 1 COMMENTS:

The manuscript presents the results of a study aimed at comparing laboratory- and non-laboratory-based WHO and GLOBORISK cardiovascular risk scores, assess their concordance, and examine relationships with sub-clinical atherosclerosis in a rural Indian cohort including 2465 adults aged 47-74 years. The study analysed a cross-section of the cohort and used appropriate, established, statistical methods to assess categorical and continuous agreement and the linear association between the 4 risk scores and various sub-clinical atherosclerosis markers. The introductory section clearly describes the rationale, aims and objectives of the study and its potential usefulness in a public health perspective. The methods section describes with sufficient detail the data collection and measurement procedures and the analytical approach. The results are clearly presented and contextualised in the Discussion section, and the conclusion are supported by the presented results. Major strengths and limitations of the study are discussed. I have listed below some specific comments that I hope the Authors would consider addressing.

Major comments

1) Discussion

While I understand the focus of the study is the comparison between laboratory and non-laboratory risk scores (rather than the comparison between risk scores), I am still surprised by the absence of comments regarding the large discrepancies between the risk estimates between WHO and Globorisk (either lab or non-lab). I wonder if it is possible to set aside some consideration regarding these differences when discussing – as the Authors do – the public health implications of their finding, by stating “Consistent with previous research, our results support the broader applicability of these models, particularly in settings with limited laboratory resources or acc

---

## [Decision Letter · Decision Letter 1]

25 Jan 2026

COMPARISON OF LABORATORY-BASED AND NON-LABORATORY-BASED WHO AND GLOBORISK CVD RISK SCORES: A CROSS-SECTIONAL ANALYSIS OF THE APCAPS COHORT

PONE-D-25-38253R1

Dear Dr. Mahajan,

We’re pleased to inform you that your manuscript has been judged scientifically suitable for publication and will be formally accepted for publication once it meets all outstanding technical requirements.

Kind regards,

Satish G Patil, PhD

Academic Editor

PLOS One

Additional Editor Comments (optional):

Reviewers' comments:

Reviewer's Responses to Questions

**Comments to the Author**

Reviewer #2: All comments have been addressed

2. Is the manuscript technically sound, and do the data support the conclusions?

Reviewer #2: Yes

3. Has the statistical analysis been performed appropriately and rigorously?

Reviewer #2: Yes

4. Have the authors made all data underlying the findings in their manuscript fully available?

Reviewer #2: Yes

5. Is the manuscript presented in an intelligible fashion and written in standard English?

Reviewer #2: Yes

Reviewer #2: All comments have been addressed and changes made to manuscript. This manuscript adds to the growing evidence of the use of non-laboratory based CVD risk prediction models in low resource settings in selected populations.

**Do you want your identity to be public for this peer review?** For information about this choice, including consent withdrawal, please see our Privacy Policy

Reviewer #2: No

---

## [Editor Report · Acceptance letter]

PONE-D-25-38253R1

PLOS One

Dear Dr. Mahajan,

I'm pleased to inform you that your manuscript has been deemed suitable for publication in PLOS One. Congratulations! Your manuscript is now being handed over to our production team.

Kind regards,

on behalf of

Prof. Dr. Satish G Patil

Academic Editor

PLOS One